# Secular Contrasts in Physical Fitness and Athletic Skills in Japanese Elementary School Students (11-Year-Olds)

**DOI:** 10.3390/ijerph21070951

**Published:** 2024-07-20

**Authors:** Yukitomo Yogi, Yasunari Ishikawa, Shuichi Takahashi

**Affiliations:** 1Department of Sports Sociology and Health Sciences, Faculty of Sociology, Kyoto Sangyo University, Kyoto 603-8555, Japan; 2Major in Physical Education, Faculty of Education, Saitama University, Saitama 338-8570, Japan; yasunari0301@mail.saitama-u.ac.jp; 3Department of Sports Science, Faculty of Physical Education, Japan Women’s College of Physical Education, Tokyo 157-8565, Japan; takahashi.shuichi@jwcpe.ac.jp

**Keywords:** physical fitness, athletic skills, secular contrast in physical fitness, secular trend in physical fitness level, declining fitness level

## Abstract

Since 1964, Japan has been conducting a wide fitness and athletic skills evaluation among 2000–3000 school children. This study used data made public by the Japan Sports Agency from four events that have continuously been evaluated from 1964 to 2021—grip strength, 50 meter dash, repetitive side jumps, and softball throw—to depict a secular trend over the course of 58 years. This is to statistically categorize these into generations, establishing secular contrast by identifying high and low scores for each event within the categorized generations, with the intent to examine the factors embedded within the background. The generations that had the highest average across all four events were the measurements made in 1970–1993 for males and in 1972–1994 for females. Changes made to the curriculum guidelines may have played a role in the differences found within each event. The goal of physical education and its syllabus and assigned hours in the curriculum guidelines are revised approximately every 10 years to meet social demand. Changes in the social and living environments surrounding children may be exhibited in the form of secular contrast in the measured physical fitness and athletic skills.

## 1. Introduction

Since 1964, the Japanese Ministry of Education, Culture, Sports, Science and Technology (MEXT) has conducted an annual evaluation of physical fitness and athletic skills to track the physical fitness and skill level of its citizens and to obtain basic information for the development of physical education and sports instruction, as well as for administrative purposes [1]. Previous studies have used the results of this evaluation to examine secular trends in children’s physical fitness and athletic skills [2,3,4]. Nishijima et al. (2003) examined the physical fitness and athletic skills of 12–17-year-olds using 34 years of data from 1964 to 1997. Their analysis revealed an improvement from 1964 to 1980, followed by a decline from 1985 [2]. Noi et al. (2002) also used 34 years of data, from 1965 to 1998, to identify the same trend as Nishijima et al., and they reported that the revisions of curriculum guidelines approximately every 10 years have an impact on changes in youth physical fitness and athletic skills [3]. The curriculum guidelines given by MEXT, based on the School Education Law, provide instruction for creating a physical education curriculum at each school that can ensure a certain basic level regardless of location [5]. 

For example, in response to the results of the long-term decline in physical fitness and athletic skills that has taken place since the late 1980s, the 1998 revision introduced new content in physical education called “body strengthening exercises” to improve physical fitness [6]. Children’s physical fitness and athletic skills are dependent on the amount of physical activity that they perform [7,8]; for this reason, we must identify secular trends, using data from 1964 to the present, to examine the factors affecting change in children’s physical fitness and athletic skills. 

To examine trends in these areas, some studies have categorized periods into 5- or 10-year chunks and discussed factors within these groups [9,10], while others have statistically categorized them into generations by examining the secular trend and identifying the contrast between high and low physical fitness and athletic skills for each generation [4,11]. For instance, Yogi et al. (2015) used data from 1964 to 2012 to clarify the factors affecting change in children’s physical fitness and athletic skills [11]. However, there have been no studies identifying the secular contrast affecting change in children’s physical fitness and athletic skills, examining the impact of the 2017 revision of the curriculum guidelines or the negative impact of COVID-19. 

This study, therefore, used 58 years of public data from 1964 to 2021, made available by the Japan Sports Agency (JSA), to statistically categorize them into generations and clarify the secular contrast by identifying the contrast of high and low fitness and skills using the categorized generations. Furthermore, the study examined the factors for these contrasts, embedded within the background of the generations in relation to the social and living environments.

## 2. Materials and Methods

### 2.1. Physical Fitness and Athletic Skill Events and Participants

Physical fitness data from 1964 to 2021 that have been made publicly available by the JSA were used. From 1964 to 1998, the physical fitness and athletic skills evaluation included 12 events: grip strength, standing forward bend, 50 meter dash, long jump, repetitive side jumps, softball throw, long-distance run, step exercise, vertical jump, back strength, prone back bend, and pull-ups. Beginning in 1999, however, changes in physique and advances in sports medicine prompted a revision of this list of events into eight—grip strength, sit-ups, sitting forward bend, 50 meter dash, standing broad jump, repetitive side jumps, softball throw, and long-distance run or a 20 meter shuttle run [12].

This study examined the four events that have been continuously evaluated from 1964—grip strength, 50 meter dash, repetitive side jumps, and softball throw.

MEXT [13] considers grip strength and repetitive side jumps to be measures of physical fitness. Grip strength is a measure of muscular strength. The speed of repetitive side jumps measures agility. The 50 meter dash and softball throw are measures of “athletic skills”. The speed of a 50 meter dash is taken as a measure of running skill. The strength and timing of a softball throw are a measure of throwing skills.

Currently, elementary school participants in the fitness and athletic skills evaluations are aged 7, 9, and 11; however, 7- and 9-year-olds were only included in the evaluation from 1998. This leaves 11-year-olds as the only participants with accumulated data (sample numbers, averages, and standard deviations) from 1964. Therefore, this study chose 11-year-olds as its target of study.

### 2.2. Extracting Participants

The boards of education in each of the 47 prefectures in Japan divided the total amount of school codes used in the basic school evaluation (basic statistical evaluation) by 3 (number of evaluated schools) to represent the sampling interval in a whole number (round down). Then, a number smaller than the sampling interval was randomly selected as the first extracted number. The value of the sampling interval was sequentially added to determine the extracted number for all three evaluated schools to extract three public schools with school codes corresponding to the extracted numbers. Where the extracted school lacks the number of reported samples, the shortage was supplemented by the school with the next number. From the extracted schools, a normal class that meets the sampling requirement was extracted from the first student in each grade, so that all students in the class (except for those deemed unable to take the test) were included. Following the evaluation, only a number of participants matching the number of reports were extracted in order, from number 1, male and female students separately, according to the student list [1]. The number of participants in the fitness and athletic skills evaluation ranged from 2000 to 3000 participants.

### 2.3. Analysis

This study, referring to the analysis method of Yogi et al. (2015) [11], calculated a standardized score (Z-score) for each of four events (grip strength, 50 meter dash, repetitive side jumps, and softball throw) using 58 years of data (1964–2021) and showed the secular change in a graph. Standardization was conducted using the following equation:Z = (X − X)/SD(1)

[X: Average for the selected period, X: Average score for each event by generation from 1964 to 2021, and SD: Standard deviation for the selected period].

The categorization of the generations was conducted by exploring cluster analysis, with the values for four events standardized by each event. In all, 58 cases were used, and the scores for the four events were the variables. The clusters were generated using Ward’s method and Euclidean squared distance. The number of clusters was set from 2 to 9, and the number of clusters that showed the most decipherable change was selected after considering its patterns and generations.

To examine the contrast between high and low fitness and skills in each event, differing by categorized generations, a two-way ANOVA was conducted using the four events and generations (clusters). When a significant interaction was found, a simple main-effects test and Tukey’s multiple comparison test were conducted. Microsoft Excel 2016 and IBM SPSS Statistics Version 25 were used for the statistical analysis of the study. The statistical significance was set at *p* < 0.05.

## 3. Results

### 3.1. Secular Trends in Fitness

Figure 1 shows the secular trend in fitness among male students. As can be seen, grip strength shows a significantly improving trend from the beginning of the evaluation to the late 1970s. Grip strength maintained a high level in the 1980s and then began a long-term trend of decline from the 1990s that continues until the present. The results for the 50 meter dash remained at a high level from 1965 to the late 1980s; however, it showed a significant decline from the early 1990s to the early 2000s, followed by a trend of gradual improvement. In the repetitive side jumps, an improving trend can be seen from the beginning of the evaluation to the mid-1980s, followed by a stagnating trend from the late 1980s to the late 1990s. A trend of significant improvement was seen from 2000. The softball throw remains at a high level from the beginning of the evaluation to the 1980s; however, a significant trend of decline is seen from the mid-1980s to 2000.

Figure 2 shows the secular trend in fitness in female students. It can be seen that grip strength shows a significant trend of improvement from the beginning of the evaluation to the late 1970s. This fluctuated at a high level from the 1980s to the early 1990s and shows a stagnating trend since that time. The 50 meter dash remained stagnant from the late 1960s to mid-1970s, followed by an improving trend from the late 1970s to early 1990s; however, a significant declining trend begins from the mid-1990s to 2000, followed by a gradually improving trend. In the repetitive side jumps, an improving trend can be seen from the beginning of the evaluation to the mid-1980s, followed by a declining trend from the late 1980s to the late 1990s, followed by significant improving trend until the present. The softball throw data show an improving trend from the beginning of the evaluation to the early 1970s and remain at a high level from the mid-1970 to the mid-1980s; however, the level begins to fall from the late 1980s to the present. 

### 3.2. Generational Categories

Figure 3 and Figure 4 show the results of the cluster analysis by gender. Four generations (clusters) emerged out of the analysis for males and females. The four generations are shown in simplified form in Figure 1 and Figure 2. Male students were grouped as follows: 1964–1969 (cluster 1), 1970–1993 (2), 1994–2009 (3), and 2010–2021 (4).

Female students were categorized into the following four clusters: 1964–1971 (cluster 1), 1972–1994 (2), 1995–2002 (3), and 2003–2021 (4)

### 3.3. Examining Secular Contrast

The results of a two-way ANOVA using events and generations as factors are shown in Table 1 and Table 2, listed by gender. Significant interactions were found for both male and female students (male Fi (interaction) = 61.41; *p* < 0.05 *, female Fi = 47.69; *p* < 0.05 *). Following a simple main-effects test and Tukey’s multiple comparison test, differences in skill level by event within and between generational clusters were found, identifying a high and low contrast in each event for the four clusters.

For male students (Table 1) in cluster 1 (1964–1969), the 50 meter dash and softball throw were higher than grip strength and repetitive side jumps. Relative to other generational clusters, the softball throw in this cluster had the highest result of all clusters except cluster 2, and the 50 meter dash was higher than that for cluster 3.

In cluster 2 (1970–1993), grip strength, 50 meter dash, and softball throw were higher than repetitive side jumps. Grip strength and softball throw were higher than those in clusters 1, 3, and 4, and the 50 meter dash was higher than that in clusters 3 and 4, while repetitive side jumps were higher than in cluster 1.

In cluster 3 (1994–2009), scores for repetitive side jumps were higher than grip strength, 50 meter dash, and softball throw. Grip strength was greater than clusters 1 and 4, and scores for repetitive side jumps were higher than those for clusters 1 and 2, and for softball throw, the scores were higher than those for cluster 4.

Within cluster 4 (2010–2021), repetitive side jump scores were higher than grip strength, 50 meter dash, and softball, while the scores for 50 meter dash were higher than those for grip and softball throw. Compared to other clusters, repetitive side jumps were higher than any other cluster. The scores for the 50 meter dash were higher than those for cluster 3.

Cluster 2 (1970–1993) had the highest average score for all events.

For female students (Table 2) in cluster 1 (1964–1971), the scores for the 50 meter dash and the softball throw were higher than those for grip strength and repetitive side jumps. Compared to other generational clusters, the softball throw was higher than all except cluster 2, and the 50 meter dash score was higher than that of cluster 3, mirroring the results found for the male students.

Within cluster 2 (1972–1994), grip strength, 50 meter dash speed, and softball throw were higher than repetitive side jumps. Compared to other clusters, the 50 meter dash values were higher than those of any other generation, grip strength was higher than for clusters 1 and 4, and softball throw was higher than that for clusters 3 and 4.

For cluster 3 (1995–2002), the grip strength was higher than 50 meter dash, repetitive side jumps, and softball throw. Compared to other clusters, grip strength was higher than that in cluster 1.

For cluster 4 (2003–2021), repetitive side jumps were higher than grip strength, 50 meter dash, and softball, while grip strength and 50 meter dash were higher than softball throw. Compared to other clusters, repetitive side jumps scores were higher those of any other cluster, just as was seen for male students. Grip strength was higher than cluster 1.

Cluster 2 (1972–1994) had the highest average for each event between generations. In addition, cluster 4 (2003–2021) was higher than clusters 1 and 3, which was different from the results for the male students.

## 4. Discussion

This study used 58 years of public data from 1964 to 2021 that have been made available by JSA, and identified trends in physical fitness and athletic skills for 11-year-old students in Japan (Figure 1 and Figure 2). Following a cluster analysis, four generations were identified for both male and female students (Figure 3 and Figure 4). The results of a two-way ANOVA and a multiple comparison test using four events as factors (grip strength, 50 meter dash, repetitive side jumps, and softball throw) alongside four categorized generations (clusters) revealed a contrast of high and low scores for each, and the average for all four events in each generation (Table 1 and Table 2). The factors of the secular trend in fitness, the categorized generations, and their contrasts will be discussed.

Cluster 1 (male: 1964–1969; female: 1964–1971) showed a period of improvement for both male and female students, as can be observed from the trend of the average scores for the four events (Figure 1 and Figure 2). The curriculum guidelines were a contributing factor, as they placed importance on the improvement of physical fitness during that period [14,15]. Elementary students spend most of their active hours at school, except during holiday breaks and weekends. Therefore, attending classes designed to improve physical fitness could very well be a factor. The scores for the softball throw were especially high during this period (Table 1 and Table 2). One contributing factor may be reported to be the time spent with ball activities relative to other sports activities, as assigned in the curriculum guidelines for that period for middle grades (8- and 9-year-olds) and higher grades (10- and 11-year-olds) [11]. Sports facilities were insufficient in Japan during that time [16], so children largely chose to play games outdoors or on the streets (often ball games) [17]. These social and living environments may have contributed to children’s throwing skills.

Cluster 2 (male: 1970–1993; female: 1972–1994) showed a period of high scores for both male and female students, as can be observed from the trend of the average scores of four events (Figure 1 and Figure 2). A multiple comparison test between generational clusters revealed that children in this period tended to be the most fit and athletically skilled overall (Table 1 and Table 2). In Japan, the construction of sports facilities began to increase around 1950 [18], but it was not until the 1970s that it increased substantially [19]. The curriculum guidelines during this period [20] continued to value fitness improvement as the course goal, as in the version for 1958 [14]. In addition, between-class physical education was introduced to embody this goal. This approach was developed to find a way to improve fitness between class periods [11]. Another approach was taken before school, including jump ropes, running, and bar exercises. These environmental factors must have relatively impacted physical fitness and athletic skills.

Cluster 3 (male: 1994–2009; female: 1995–2002) showed a decline for both male and female students, as observed from the trend of the average scores for the four events (Figure 1 and Figure 2). A significant period of decline is observable from the early 1990s to 2000. The revised curriculum guidelines in 1977 may have been a contributing factor, as they showed a changed goal for the physical education course [21]. The previous goal, to improve fitness, had been replaced by the goal of enjoying physical education [22]. At this time, sports had come to be recognized as an important element in society and culture, and it needed to be enjoyed throughout one’s lifetime [23]. As this idea began to take root, the goal of curriculum guidelines took a turn, thereby drastically changing the content of the classes. Physical education tasks created to improve fitness were replaced with others to encourage students to enjoy sports. A teacher-led, uniform, and simultaneous class changed to a student-led, enjoyment-oriented class, ultimately decreasing the amount of exercise that children participated in and resulting in a decline in children’s physical fitness and athletic skills, which created an impact on this generation after a lapse of time. This decline was significantly evident in female students. Adolescent girls generally had a lower level of physical activity and did not prefer exercise, unlike boys [24,25]. These differences in characteristics may have influenced this result.

Cluster 4 (male: 2010–2021; female: 2003–2021) showed stagnation for both male and female students, as observed from the trend of the average scores of four events (Figure 1 and Figure 2). This period can be understood as one for which a declining trend was seen, as in cluster 3. The curriculum guidelines revised in 1998 may have been a contributing factor, as these introduced new content called a “body strengthening exercise.” To improve the declining trend in physical fitness that had begun in the 1980s, body strengthening exercises were implemented that were designed to improve the level of fitness and relieve tension [26]. This was retained in the 2008 revision [27]. The adoption of this content to improve fitness may have helped stop the declining trend in physical fitness. On the other hand, examining the secular trend by events, repetitive side jumps significantly improved from 2000 for both male and female students. Repetitive side jumps are used to assess speed (agility) in a repeated sidestep motion. In Japan, the number of local playfields and sports facilities increased from the 1970s to around 2010 [28]. This led to an increase in the number of sports clubs and elementary students engaging in local sports activities after school and on the weekends. The outdoor sports of baseball and soccer and the indoor sports of basketball, volleyball, and badminton are popular in Japan [29]. On the other hand, activities related to the enjoyment of sports also continued in the physical education classes. Children are spending much more time playing games. Such conditional factors may have contributed to improving agility in elementary students.

Children’s social and living environments necessarily have an impact on their physical fitness and athletic skills. The curriculum guidelines that are revised approximately every 10 years to meet social demand and change must have a substantial impact on children’s social environment. In Western countries, physical education classes’ hours and breaks have been reduced to improve academic grades, and elementary students have fewer opportunities for physical activities [30,31,32,33]. In Japan, the introduction of a pressure-free education, or so-called *yutori* education, in the 1998 revision of the curriculum guidelines, decreased the number of hours given for physical education. This revision may have been a factor in the decline in physical education [11]. The COVID-19 pandemic has also contributed to a rapid decline in physical education. To improve the physical fitness of children, the future bearers of society, it is important to build a society that is full of life and energy. In their living environment, children need to have a maintained space where they can easily play and exercise near their homes. Although playing fields and sports facilities have increased, the effects of the aging society have led to less playground equipment being geared toward children. An increased number of parks are prohibiting playing with a ball, such as playing catch or soccer. A polarization between children who exercise and do not exercise at all has been reported as a problem contributing to a decline in physical fitness [34,35]. Maintaining an environment that allows children to easily play in a park near to their living area is necessary. While this study targeted elementary school students (11-year-olds), the JSA has data on 13- and 16-year-olds. The amount of physical activity that children participate in is reported to decline with age [36,37,38]. Research on the secular contrast in middle school students (13-year-olds) and high school students (16-year-olds) will be conducted in the future.

## 5. Conclusions

Using the data regarding physical fitness and athletic skills made available by the JSA (1964–2021), this study identified secular trends for grip strength, 50 meter dash, repetitive side jumps, softball throws, and the average of these four events for elementary students (11-year-olds) in Japan. An exploratory cluster analysis using the scores of four events, conducted to determine generations, resulted in categorizing the period into four generations separately for both male and female students. Next, a two-way ANOVA was performed to examine the difference between events within and between the four generations, revealing a high and low score contrast, proving the secular contrast. The contributing factors to the contrast between events may be the changes made to the curriculum guidelines as well as to social and living environments across generations. Encouraging children to feel joy and satisfaction in physical activities as well as increasing their motivation for physical education classes is necessary for improving children’s fitness. In this generation, it is necessary to increase motivation in physical education classes that will lead to physical activities after school and on the weekends, ultimately increasing the amount of exercise and improving physical fitness and athletic skill. 

This study was thus conducted using 58 years of public data from 1964 to 2021, made available by JSA. It can therefore be asserted that the research results are representative to a certain extent; however, the limitations of the study need to be noted, namely that the evolution of measuring instruments (grip strength meters and stopwatches), shoes, and the increase in children’s physique over the past 58 years have affected the results of this study.

## Figures and Tables

**Figure 1 ijerph-21-00951-f001:**
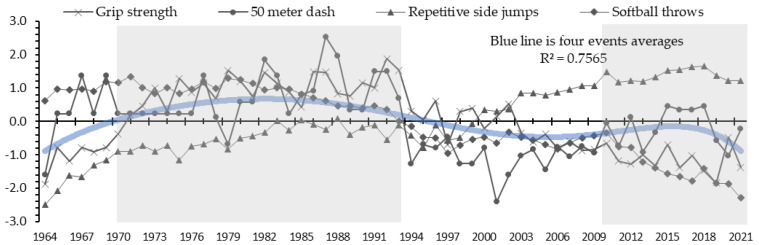
Secular fitness trend in male students (1964–2021).

**Figure 2 ijerph-21-00951-f002:**
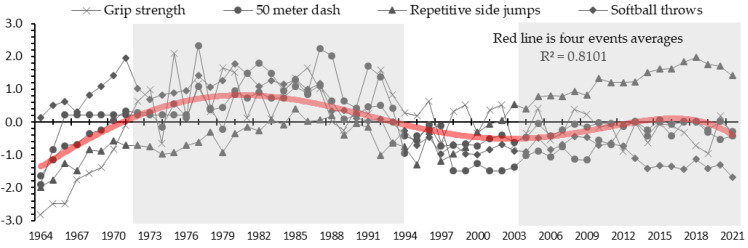
Secular fitness trend in female students (1964–2021).

**Figure 3 ijerph-21-00951-f003:**
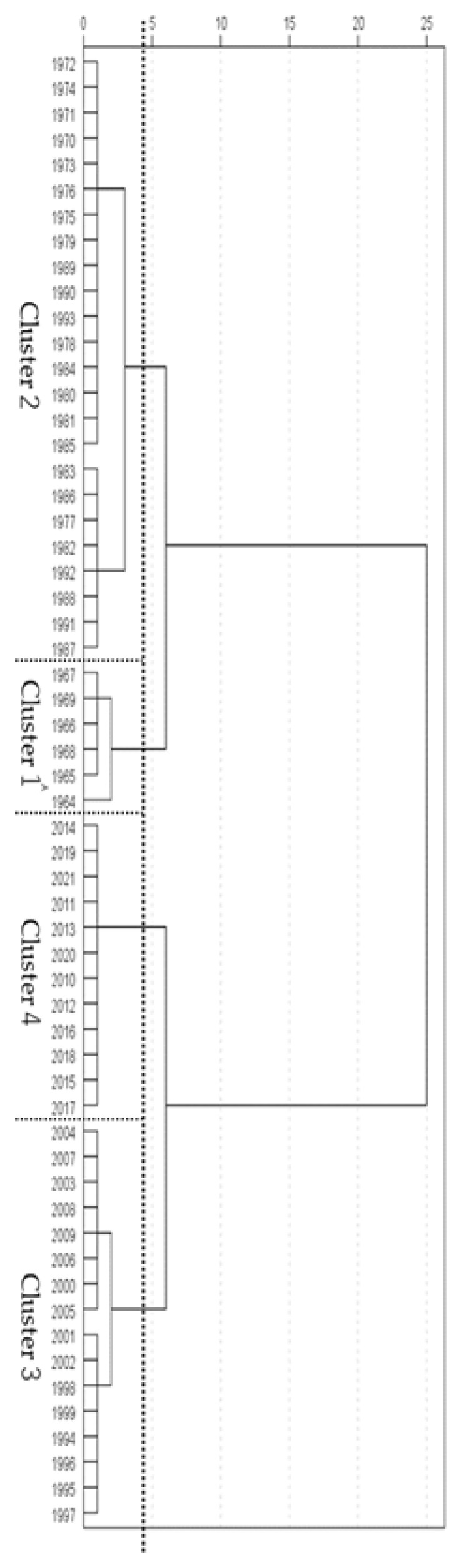
Results of cluster analysis (male students).

**Figure 4 ijerph-21-00951-f004:**
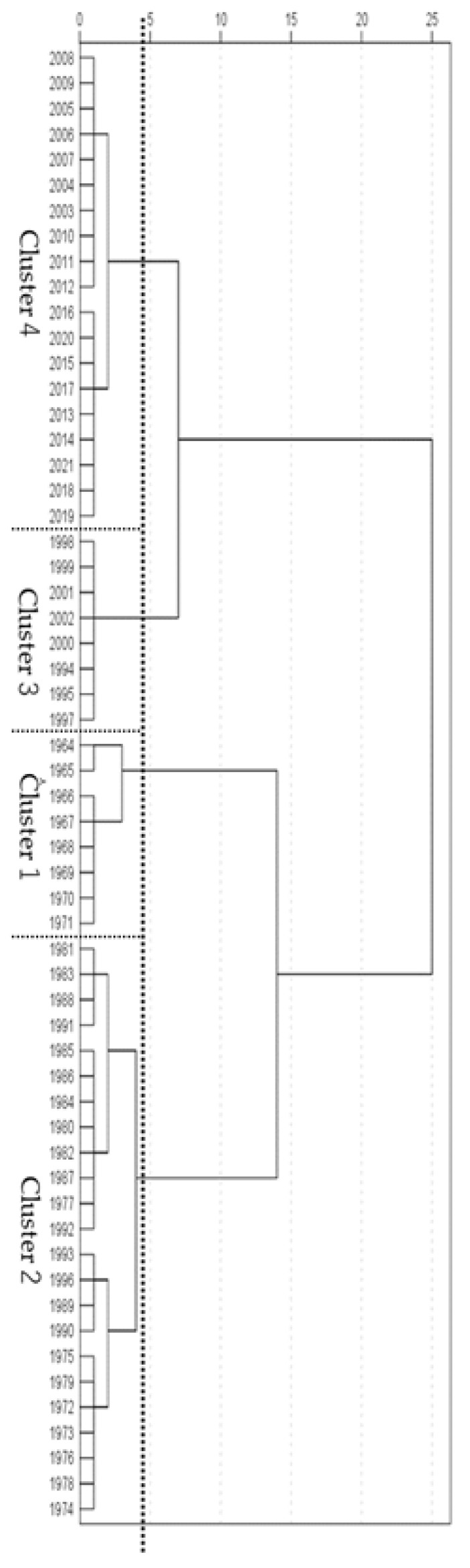
Results of cluster analysis (female students).

**Table 1 ijerph-21-00951-t001:** Generational cluster variables for each event (averages and standard deviations) and results of two-way analysis of variance and multiple comparison test (male students).

	4 Generational Clusters								F Value (*p* < 0.05 *)	Contrasts of 4 Generational Clusters (Multiple Comparison Test, *p* < 0.05)				
	Cluster 1 (1964–1969)		Cluster 2 (1970–1993)		Cluster 3 (1994–2009)		Cluster 4 (2010–2021)			Event Significant Differences within Generational Clusters				Event Significant Differences between Generation Clusters and Four Events Averages
Events	Averages	SD	Averages	SD	Averages	SD	Averages	SD		Cluster 1	Cluster 2	Cluster 3	Cluster 4	
➀ Grip strength	−1.06	0.44	0.96	0.51	−0.19	0.52	−1.14	0.39	Fe = 3.96 * Fg = 48.12 * Fi = 61.41 *	➀ < ➁・➃	➀ > ➂	➁ < ➀ < ➂	➀ < ➁・➂	2 > 1・3・4, 3 > 1・4
➁ 50 meter Dash	0.31	1.09	0.73	0.74	−1.09	0.47	−0.18	0.54		➁ > ➀・➂	➁ > ➂	➁ < ➀・➂・➃	➂ > ➁ > ➀・➃	2 > 3・4, 1 > 3, 4 > 3
➂ Repetitive side jumps	−1.72	0.49	−0.47	0.35	0.31	0.64	1.38	0.18		➂ < ➁・➃	➂ < ➀・➁・➃	➂ > ➀・➁・➃	➂ > ➀・➁・➃	4 > 1・2・3, 3 > 1・2, 2 > 1
➃ Softball throws	0.94	0.18	0.84	0.34	−0.56	0.19	−1.41	0.56		➃ > ➀・➂	➃ > ➂	➁ < ➃ < ➂	➃ < ➁・➂	1 > 3・4, 2 > 3・4, 3 > 4
Four events averages	−0.38	0.52	0.52	0.28	−0.38	0.14	−0.34	0.25		－	－	－	－	2 > 1・3・4

F value; Fe (event), Fg (generational), Fi (interaction), *p* < 0.05 *.

**Table 2 ijerph-21-00951-t002:** Generational cluster variables for each event (averages and standard deviations) and results of two-way analysis of variance and multiple comparison test (female students).

	4 Generational Clusters								F Value (*p* < 0.05 *)	Contrasts of 4 Generational Clusters (Multiple Comparison Test, *p* < 0.05)				
	Cluster 1 (1964–1971)		Cluster 2 (1972–1994)		Cluster 3 (1995–2002)		Cluster 4 (2003–2021)			Event Significant Differences within Generational Clusters				Event Significant Differences between Generation Clusters and Four Events Averages
Events	Averages	SD	Averages	SD	Averages	SD	Averages	SD		Cluster 1	Cluster 2	Cluster 3	Cluster 4	
➀ Grip strength	−1.75	0.79	0.78	0.66	0.15	0.42	−0.27	0.41	Fe = 4.63 * Fg = 40.01 * Fi = 47.69 *	➀ < ➁・➃	➀ > ➂	➀ > ➁・➂・➃	➂ > ➀ > ➃	2 > 1・4, 3 > 1, 4 > 1
➁ 50 meter Dash	−0.18	0.79	0.91	0.77	−1.08	0.55	−0.57	0.44		➁ > ➀・➂	➁ > ➂	➁ < ➀	➂ > ➁ > ➃	2 > 1・3・4, 1 > 3
➂ Repetitive side jumps	−1.22	0.49	−0.36	0.39	−0.66	0.50	1.23	0.46		➂ < ➁・➃	➂ < ➀・➁・➃	➂ < ➀	➂ > ➀・➁・➃	4 > 1・2・3, 2 > 1
➃ Softball throws	0.72	0.42	0.84	0.67	−0.76	0.24	−1.00	0.38		➃ > ➀・➂	➃ > ➀	➃ < ➀	➃ < ➀・➁・➂	2 > 3・4, 1 > 3・4
Four events averages	−0.61	0.56	0.54	0.36	−0.59	0.14	−0.15	0.19		－	－	－	－	2 > 1・3・4, 4 > 1・3

F value; Fe (event), Fg (generational), Fi (interaction), *p* < 0.05 *.

## Data Availability

Data are available from the corresponding author upon reasonable request.

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
