# Peer review of "Secular Contrasts in Physical Fitness and Athletic Skills in Japanese Elementary School Students (11-Year-Olds)"

_ijerph, 2024, doi:10.3390/ijerph21070951_

Round 1
Reviewer 1 Report
Comments and Suggestions for Authors
Lines 29-37. The introductory sentence begins with COVID-19. The purpose of the study is not to examine a COVID-related process. It should start with the scope of the study.
Line 37. The sentence should start with a different structure instead of, however.
Line 64. The unique value (originality) of the study should be better explained. What makes the study important? This question should be answered.
Line 43. Reference 15 on MEXT, line 81: Reference 26 on MEXT should be checked. If they are different, there is no problem.
Line 107. Statistical analysis was used correctly.
Line 223: Is there a case of elementary school students being contracted?
The limitation of the study is that the method used in 1964 is the same as the method used in 2021. How were the measurement methods standardised? This should be explained.
The subject and content of the study are original. Its long duration makes it valuable. It can be published after corrections.
Comments on the Quality of English LanguageMinor editing of English language required
Author Response
Comments 1: Lines 29-37. The introductory sentence begins with COVID-19. The purpose of the study is not to examine a COVID-related process. It should start with the scope of the study.
Responses 1: As we found this COVID-related introductory text to be unnecessary, we deleted the first paragraph of the Introduction.
Comments 2: Line 37. The sentence should start with a different structure instead of, however.
Responses 2: In relation to the deletion associated with the above comment (Lines 29-37), we have also deleted the sentence that contains this “however”. (However, children’s physical fitness had already begun to decline before the impacts of the pandemic arrived [13,14])
Comments 3: Line 64. The unique value (originality) of the study should be better explained. What makes the study important? This question should be answered.
Responses 3: We have added previous research and conjunctions to clarify the originality of our study. (Lines 51-59)
Comments4: Line 43. Reference 15 on MEXT, line 81: Reference 26 on MEXT should be checked. If they are different, there is no problem.
Responses 4: We have checked the references again and confirmed that these references are correct. As a side note, the Ministry of Education, Culture, Sports, Science and Technology (MEXT) oversees the administration of education in Japan, but as of October 2015, the administration of physical education and sports was transferred to the Japan Sports Agency(JSA).
Comments5: Line 107. Statistical analysis was used correctly.
Responses 5: To make it clear that we have done the statistical process correctly, we have added a reference. (line 105)
Comments6: Line 223: Is there a case of elementary school students being contracted?
Responses 6: We have thought it over and could not understand the meaning of this question. We sincerely would like to answer this reviewer's question and would appreciate it if he/she could more specifically explain this question to us.
Comments7: The limitation of the study is that the method used in 1964 is the same as the method used in 2021. How were the measurement methods standardised? This should be explained.
Responses 7: We have added the limitations of the study to the last paragraph of the conclusion. (Lines 316-321)
Comments8: The subject and content of the study are original. Its long duration makes it valuable. It can be published after corrections.
Responses 8: We appreciate your compliments. We hope that this revision answers your question.
Reviewer 2 Report
Comments and Suggestions for Authors
Monitoring fitness tests is crucial in aligning school curricula with the evolving needs of children. Fitness assessments provide valuable insights into the physical development and health status of students, which can inform educators and policymakers in designing effective physical education programs. By tracking trends in fitness and athletic skills over time, we can identify areas of improvement and ensure that the curriculum promotes overall well-being and physical literacy among students. These assessments also help in recognizing the impact of social and environmental changes on children's fitness, enabling a more responsive and adaptive educational approach.
The inclusion of the extensive discussion on the impact of the COVID-19 pandemic in the introduction section is not relevant to the primary focus of this study, which aims to analyze long-term trends in physical fitness and athletic skills among Japanese elementary school students. I recommend omitting this section to maintain a more focused and pertinent introduction.
Overall, the study is well-executed and provides valuable insights into the secular trends in physical fitness and athletic skills among 11-year-old students in Japan. The use of data spanning from 1964 to 2021 is commendable and allows for a comprehensive analysis of changes over time. The statistical methods employed, including Z-scores and cluster analysis, are appropriate for identifying trends and categorizing different generations.
The findings clearly highlight significant periods of improvement and decline in various fitness metrics, which are well-supported by the data presented. The discussion on the potential impact of curriculum changes and social environments on these trends is particularly insightful and adds depth to the analysis.
Questions for the Authors: Could you streamline the introduction by omitting or shortening the discussion on the impact of the COVID-19 pandemic, given that the primary focus of the study is to analyze long-term trends in physical fitness and athletic skills among Japanese elementary school students from 1964 to 2021? Clarification on Social and Environmental Factors: Can you provide a more detailed analysis or data on how specific social and environmental changes, such as urbanization or changes in leisure activities, have influenced the trends in children's physical fitness over the studied period? Please describe the mechanisms through which changes in the physical education curriculum guidelines directly impacted the fitness trends observed in the study? Specifically, how did the shift in the curriculum in 1977 to emphasize enjoyment over fitness improvement affect students' physical activity levels and fitness outcomes?
Author Response
Questions 1: Could you streamline the introduction by omitting or shortening the discussion on the impact of the COVID-19 pandemic, given that the primary focus of the study is to analyze long-term trends in physical fitness and athletic skills among Japanese elementary school students from 1964 to 2021?
Response 1: In response to your suggestion, we have deleted the first paragraph of the Introduction and have included a short background note on line 59 regarding COVID-19.
Questions 2: Clarification on Social and Environmental Factors: Can you provide a more detailed analysis or data on how specific social and environmental changes, such as urbanization or changes in leisure activities, have influenced the trends in children's physical fitness over the studied period?
Response 2: In the last paragraph of the Discussion (Lines 278-285, 288-294), we cited previous studies in Japan to describe how changes in social and living environments may have affected the trend in physical fitness, and we have kept this scope in this study. However, as you pointed out, in the future we would like to examine the relationship with physical fitness data through multidimensional analysis with variables of social environment and living environment data that are assumed to be related.
Questions 3: Please describe the mechanisms through which changes in the physical education curriculum guidelines directly impacted the fitness trends observed in the study? Specifically, how did the shift in the curriculum in 1977 to emphasize enjoyment over fitness improvement affect students' physical activity levels and fitness outcomes?
Response 3: We have added text to the fifth paragraph of the Discussion as underlined below. (Line 253-254)
A teacher-led, uniform, and simultaneous class changed to a student-led, enjoyment-oriented class, ultimately decreasing the amount of exercise that children participated in and resulting in a decline in children's physical fitness and athletic skills, which created an impact on this generation after a lapse of time.